supramolecular chemistry

$CyH_2Q[6]$, 2-phenylbenzimidazole, host–guest interaction, ion recognition, limit of detection

**Author for correspondence:**
Peihua Ma
e-mail: phma@gzu.edu.cn

# Study on the host–guest complex of dicyclohexanocucurbit[6]uril and 2-phenylbenzimidazole, and its recognition effect toward $Fe^{3+}$

Jun Zheng, Lian An, Jie Gao, Lin Zhang, Xinan Yang, Weiwei Zhao, Siyuan Cheng and Peihua Ma

Key Laboratory of Macrocyclic and Supramolecular Chemistry of Guizhou Province, Guizhou University, Guiyang 550025, People's Republic of China

PM, 0000-0002-4965-0632

This paper has selected dicyclohexanocucurbit[6]uril ($CyH_2Q[6]$) as the host and 2-phenylbenzimidazole (**G**) as the guest to investigate the host–guest interaction mode between $CyH_2Q[6]$ and **G**. Under acidic conditions, the complex was characterized using nuclear magnetic resonance, ultraviolet and fluorescence spectroscopy. The results show that the molecular ratio of $CyH_2Q[6]$ to **G** is 2 : 1. The crystals were cultured with $ZnCl_2$ as a structural inducer under acidic conditions and single crystal X-ray diffraction showed that the molecular ratio of $CyH_2Q[6]$ to **G** is 1 : 3. The G@$CyH_2Q[6]$ was used as a fluorescent probe to identify metal cations. The probe exhibits a good selective recognition effect toward $Fe^{3+}$ ions, which involves a reduced fluorescence intensity with a limit of detection of $1.321 \times 10^{-6}$ mol $l^{-1}$.

## 1. Introduction

Benzimidazole compounds [1–5] are aromatic heterocyclic compounds containing two nitrogen atoms. Benzimidazole derivatives and metal complexes exhibit good biological activity. Benzimidazole and its derivatives containing an imidazole ring have important medicinal value and have important applications as anti-fungal, anti-rheumatic, deworming, analgesic, anti-inflammatory and anti-cancer agents, among others [6–13]. $Fe^{3+}$ is widely distributed in nature and is one of

**Figure 1.** The structure of (*a,b*) CyH$_2$Q[6] and (*c*) 2-phenylbenzimidazole.

the indispensable trace elements found in the human body. It plays an important role in metabolism. Cucurbit[n]uril (Q[n] or CB[n]) [14–28] is a new type of macrocyclic compound discovered after cyclodextrins, crown ethers and calixarenes. However, most ordinary cucurbit[n]uril have poor solubility, the development of cucurbit[n]urils has been significantly limited. Through the efforts of some researchers, several modified cucurbit[n]urils, such as methyl-, hydroxyl-, cyclopentyl- and cyclohexyl-substituted cucurbit[n]urils have been reported [29–35]. Dicyclohexanocucurbit[6]uril (CyH$_2$Q[6]) is a modified cucurbituril, which has made great contributions in host–guest chemistry, coordination chemistry, etc. [36–39].

Herein, CyH$_2$Q[6] was selected as the host and 2-phenylbenzimidazole (**G**) used as the guest to study the host–guest interaction (figure 1). Nuclear magnetic resonance (NMR), ultraviolet (UV) and fluorescence spectroscopy, and single crystal X-ray diffraction were used to characterize the host–guest complex. Regarding the study of cucurbit[nuril induced to sensing for metal ions many scientific researchers have made great efforts and have reported in related fields40,41]. Specifically, CyH$_2$Q[6] with 2-phenylbenzimidazole was designed as a fluorescent probe and its ability to recognize metal ions was explored using the change in its fluorescence intensity. The results show that the fluorescent probe can selectively recognize $Fe^{3+}$ cations, which exhibit a reduced fluorescence intensity and can be used to detect the $Fe^{3+}$ concentration in a sample using the reduced fluorescence effect.

# 2. Experimental section

## 2.1. Instruments and reagents

Bruker D8 Venture X-ray single crystal diffractometer (Bruker, Germany), JNM-ECZ400S/L1 400M superconducting NMR spectrometer, a UV-2700 dual-beam UV–vis spectrophotometer (Shimadzu Instruments Co. Ltd), VARIANCARYE-CLIPSE fluorescence spectrophotometer (Varian, USA). All materials were reagent grade and used without any further purification. CyH$_2$Q[6] (purity ≥97%) was prepared in the Key Laboratory of Macrocyclic and Supramolecular Chemistry of Guizhou Province, China.

## 2.2. Interaction between CyH$_2$Q[6] and 2-phenylbenzimidazole: $^1$H NMR titration

CyH$_2$Q[6] was added to deuterated water containing a certain amount of deuterated hydrochloric acid to prepare a solution with a concentration of $1.0 \times 10^{-3}$ mol l$^{-1}$ (pH = 1). Fifty microlitres of this solution and 450 µl of deuterium chloride solution was loaded into an $^1$H NMR tube and a certain amount of 2-phenylbenzimidazole (n(**G**)/n(CyH$_2$Q[6]) = 0.2, 0.4, 0.6, …) was added in sequence and the $^1$HNMR spectrum recorded at 293 K.

## 2.3. UV and fluorescence titration

CyH$_2$Q[6] and 2-phenylbenzimidazole were formulated into an acidic aqueous solution (pH = 1) with a concentration of $1.00 \times 10^{-2}$ and $1.00 \times 10^{-3}$ mol l$^{-1}$, respectively, to keep the concentration of the guest

**Table 1.** The crystallographic parameters of the complex.

| empirical formula | $C_{83}H_{82}Cl_8N_{30}O_{14}Zn_2$ | $Dc$ (g cm$^{-3}$) | 1.349 |
|---|---|---|---|
| $M_r$ | 2138.12 | F(000) | 4392 |
| crystal system | monoclinic | $\mu$ (mm$^{-1}$) | 0.729 |
| space group | $P2_1/c$ | data/params | 17 898/1187 |
| $a$ (Å) | 15.6869(8) | $R_{int}$ | 0.1002 |
| $b$ (Å) | 48.224(2) | $R[I > 2\sigma(I)]^a$ | 0.1320 |
| $c$ (Å) | 14.3174(6) | $wR[I > 2\sigma(I)]^b$ | 0.3543 |
| $\alpha$ (deg) | 90 | $R$(all data) | 0.1779 |
| $\beta$ (deg) | 103.561(2) | $wR$(all data) | 0.3808 |
| $\gamma$ (deg) | 90 | GOF ($F^2$) | 1.350 |
| $V$ [Å$^3$] | 10529.0(9) | $T$ (K) | 300.0 |
| Z | 4 | CCDC | 2 100 651 |

[a]Conventional $R$ on $Fhkl$: $\sum ||F_o| - |F_c||/\sum |F_o|$.
[b]Weighted $R$ on $|Fhkl|^2$: $\sum [w(F_o^2 - F_c^2)^2]/ \sum [w(F_o^2)^2]^{1/2}$.

unchanged and an appropriate amount of CyH$_2$Q[6] (n(G)/n(CyH$_2$Q[6]) = 0.2, 0.4, 0.6, …) was then added and analysed using UV absorption and fluorescence spectroscopy.

## 2.4. Crystal culture and testing

CyH$_2$Q[6] (10 mg, 9.05 µmol) and 2-phenylbenzimidazole (5 mg, 25.76 µmol) were dissolved in a 10 ml round bottom flask containing 5 ml of HCl (3 mol l$^{-1}$), and ZnCl$_2$ (0.67 g) was added. The resulting mixture was shaken for 3 min at room temperature and heated to reflux for 5 min with stirring. The mixture was cooled and left to stand for about one week to obtain colourless crystals. A regular and transparent crystal of the complex was selected and fixed on a glass filament using petroleum jelly for X-ray single crystal diffraction analysis on a Bruker D8 Venture X-ray single crystal diffractometer in ω-scan mode using graphite to monochromatize the Mo-K$\alpha$ rays ($\lambda$ = 0.71073 Å, $\mu$ = 0.828 mm$^{-1}$). The crystal data were collected and Lorentz polarization, and absorption correction carried out. SHELXT-14 and SHELXL-14 program packages were used for structural analysis and full matrix least-squares refinement. All non-hydrogen atoms were anisotropically refined using the analytical expression of the neutral atom scattering factor and combined with anomalous dispersion correction. The SQUEEZE program in the PLATON package was used to delete some of the disordered solvent molecules. The X-ray crystallographic data for structures reported in this study have been deposited in the Cambridge Crystallographic Data Center under accession no. CCDC: 2100651. These data can be obtained free of charge via https://www.ccdc.cam.ac.uk/data_request/cif. The crystal data and structure modification parameters are shown in table 1.

## 2.5. Selective fluorescence measurement of metal ions

First, all the metal ions studied were accurately prepared into 0.2 mol l$^{-1}$ solutions using deionized water and then, under acidic conditions (pH = 1), a solution of **G** at a concentration of $5 \times 10^{-5}$ mol l$^{-1}$ prepared with deionized water was used to form a **G**@CyH$_2$Q[6] solution. Three millilitres of the $5 \times 10^{-5}$ mol l$^{-1}$ **G**@CyH$_2$Q[6] solution was added to a quartz fluorescent cuvette and the different metal cation solutions added. The amount of metal cation added to the **G**@CyH$_2$Q[6] probe was 10 times the equimolar concentration. A fluorophotometer was used to detect the fluorescence intensity change at the maximum excitation wavelength ($\lambda_{ex}$ = 298 nm); the slit width was 5 nm/5 nm, voltage was 400 V, and scanning range was 250–600 nm.

## 2.6. Fluorescent quenching experiment of metal ions

Under acidic conditions (pH = 1), a solution of **G**@CyH$_2$Q[6] and Fe$^{3+}$ at a concentration of $5 \times 10^{-5}$ mol l$^{-1}$ was prepared with deionized water and 3 ml of the resulting solution was added to a quartz fluorescent cuvette. Different metal cation solutions were added to the solution; the amount of metal cation added

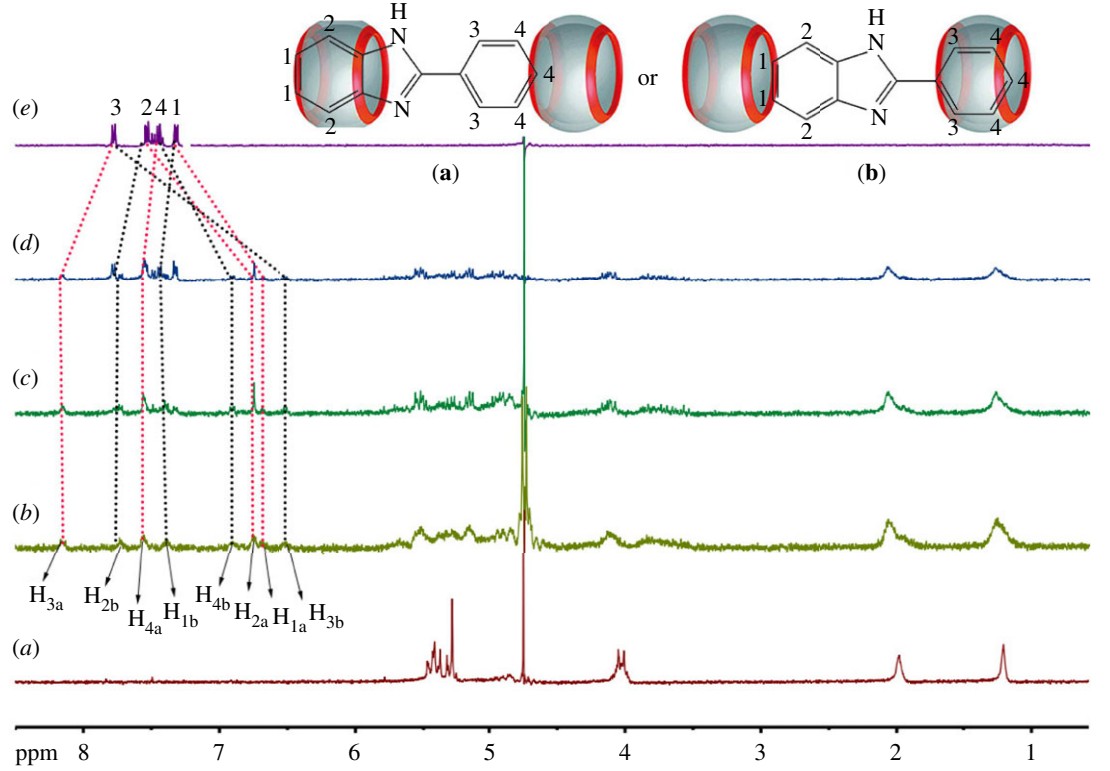

**Figure 2.** $^1$HNMR spectra (25°C, 400 MHz) of the interaction between CyH$_2$Q[6] and **G** recorded in D$_2$O upon adding (*a*) 0, (*b*) 0.4, (*c*) 0.6 and (*d*) 1.0 equivalents of **G**, and (*e*) pure **G**.

was 10 times the equimolar concentration of the **G**@CyH$_2$Q[6] and Fe$^{3+}$ solution. After mixing uniformly, a fluorophotometer was used to detect the fluorescence signal intensity within 2 min at the maximum excitation wavelength ($\lambda_{ex}$ = 298 nm); the slit width was 5 nm/5 nm, voltage was 400 V, and scanning range was 250–600 nm.

## 2.7. Titration experiment of Fe$^{3+}$

Three millilitres of a $5 \times 10^{-5}$ mol l$^{-1}$ solution of the **G**@CyH$_2$Q[6] probe was added to a quartz cuvette, followed by Fe$^{3+}$ and **G**@CyH$_2$Q[6] = 0, 0.5, 1.0, 1.5, 2.0, 2.5. After adding 3.0, 3.5, 4.0, … , 28.0 equivalent of the Fe$^{3+}$ solution to the quartz cuvette in turn and mixing uniformly, a fluorophotometer was used to detect the fluorescence signal intensity within 2 min at the maximum excitation wavelength ($\lambda_{ex}$ = 298 nm); the slit width was 5 nm/5 nm, voltage was 400 V, and scanning range was 250–600 nm.

# 3. Results and discussion

## 3.1. $^1$HNMR Titration of the host–guest interaction

Figure 2 shows the $^1$HNMR titration spectra obtained when adding different equivalents of 2-phenylbenzimidazole (**G**) to CyH$_2$Q[6]. It can be seen from the figure that, upon the addition of **G**, each proton peak of **G** splits into two groups of peaks, moving to the high field and low field, respectively. A preliminary judgement can be made on this. There are two modes of action for **G** and CyH$_2$Q[6] (shown by the black and red curves in figure 2), namely mode **a** and mode **b**. In mode **a**, the benzimidazole part enters the cavity of CyH$_2$Q[6] and the benzene ring is outside the port of CyH$_2$Q[6]; the benzimidazole part of **G** is shielded and the benzene ring is unshielded. In mode **b**, the benzene ring enters the cavity of CyH$_2$Q[6] and the benzimidazole part is outside the port of CyH$_2$Q[6]; the benzene ring part of G is shielded and the benzimidazole is unshielded. When 0.4 equivalents of **G** were added, the chemical shift values of H2a and H1a in **G** correspond to their free state moving 0.74 and 0.64 ppm to the high field, respectively; H3a and H4a move 0.37 and 0.09 ppm to the low field, respectively. The chemical shift values of H3b and H4b correspond to their free state

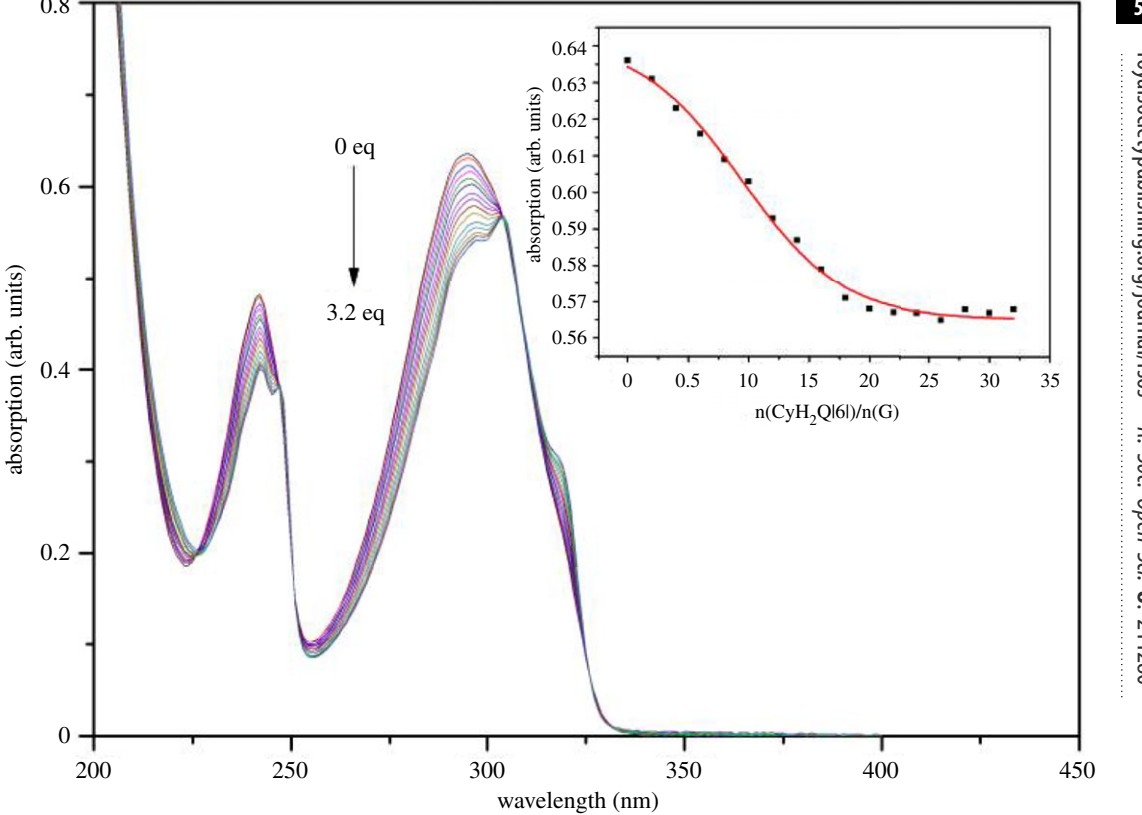

**Figure 3.** UV spectra and trend chart obtained for the solution of **G** ($5 \times 10^{-5}$ mol l$^{-1}$) upon gradually adding CyH$_2$Q[6] (0, 0.2, 0.4, 1.6, … , 3.2 equivalents).

moving 1.27 and 0.52 ppm to the high field, respectively; H2b and H1b move 0.24 and 0.07 ppm to the low field, respectively. When 0.6 equivalents of **G** were added, the chemical shift of the proton signal peaks of **G** are the same as the chemical shifts observed after adding 1.0 equivalent, and a free guest peak appears at the same time. This shows that the amount of the guest was excessive, indicating that the ratio of CyH$_2$Q[6] to **G** was 2 : 1.

## 3.2. UV absorption spectroscopy of the host–guest interaction

Figure 3 shows the UV absorption spectra obtained upon the interaction of CyH$_2$Q[6] with **G**. n(CyH$_2$Q[6])/n(G) = 0.2, 0.4, 0.6, … , 3.2, was added to the guest solution and the UV spectra recorded. As the amount of CyH$_2$Q[6] added to the guest solution increases, the UV absorption of the guest shows a regular change and gradually decreases. When the host–guest ratio was 2 : 1, the fitting curve exhibits an inflection point and as the concentration of CyH$_2$Q[6] further increases, the absorption intensity basically remains unchanged, indicating that the ratio of CyH$_2$Q[6] to **G** was 2 : 1.

## 3.3. Crystal structure

CyH$_2$Q[6] and 2-phenylbenzimidazole (**G**) were added with ZnCl$_2$ as a structural inducer and the crystal structure of the resulting complex was determined using X-single crystal diffraction. The asymmetric structural unit of the complex includes one CyH$_2$Q[6] molecule, three molecules of **G**, two [ZnCl$_4$]$^{2-}$ ions and two free H$_2$O molecules (figure 4a). Figure 4b clearly shows that **G** enters the cavity of the cucurbit[n]uril, and the two nitrogen atoms on the imidazole ring interact with the carbonyl oxygen atoms at the port of the cucurbit[n]uril via hydrogen bonding (as shown in figure 4b). Figure 4c shows the stacking diagram of the crystal structure of the complex viewed along the c-axis. The other two molecules of 2-phenylbenzimidazole act as a bridge connecting the two layers of cucurbit[n]uril, so that the complexes form a regular arrangement.

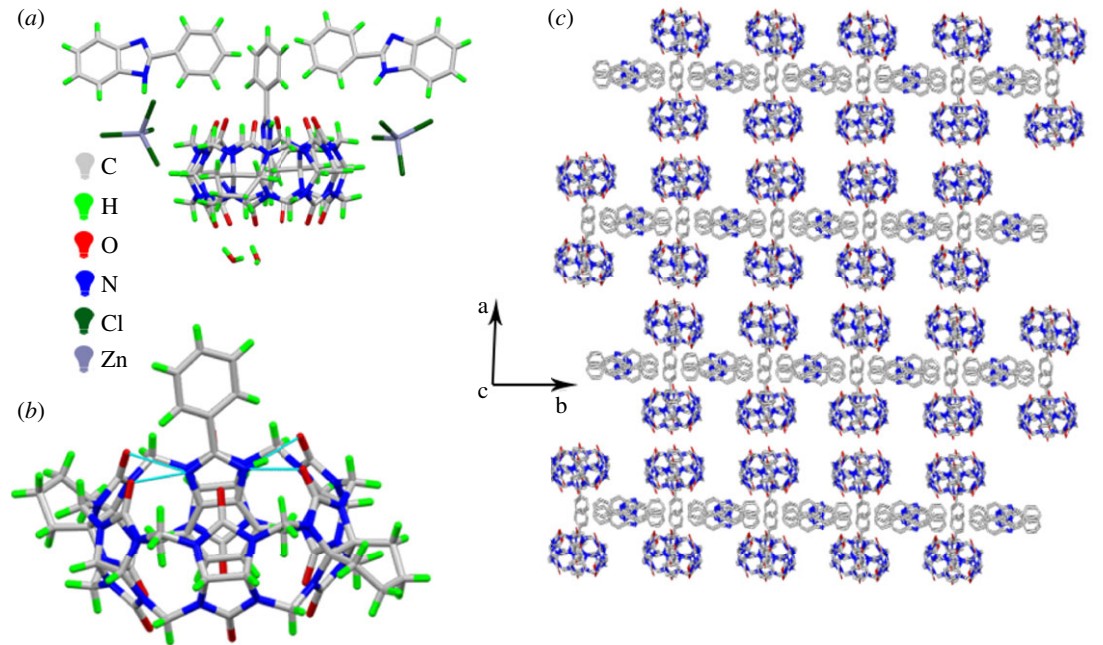

**Figure 4.** Crystal structure of CyH$_2$Q[6] and **G**: (*a*) Asymmetric structural unit, (*b*) host–guest interaction, and (*c*) stacked diagram of the structure along the c-axis.

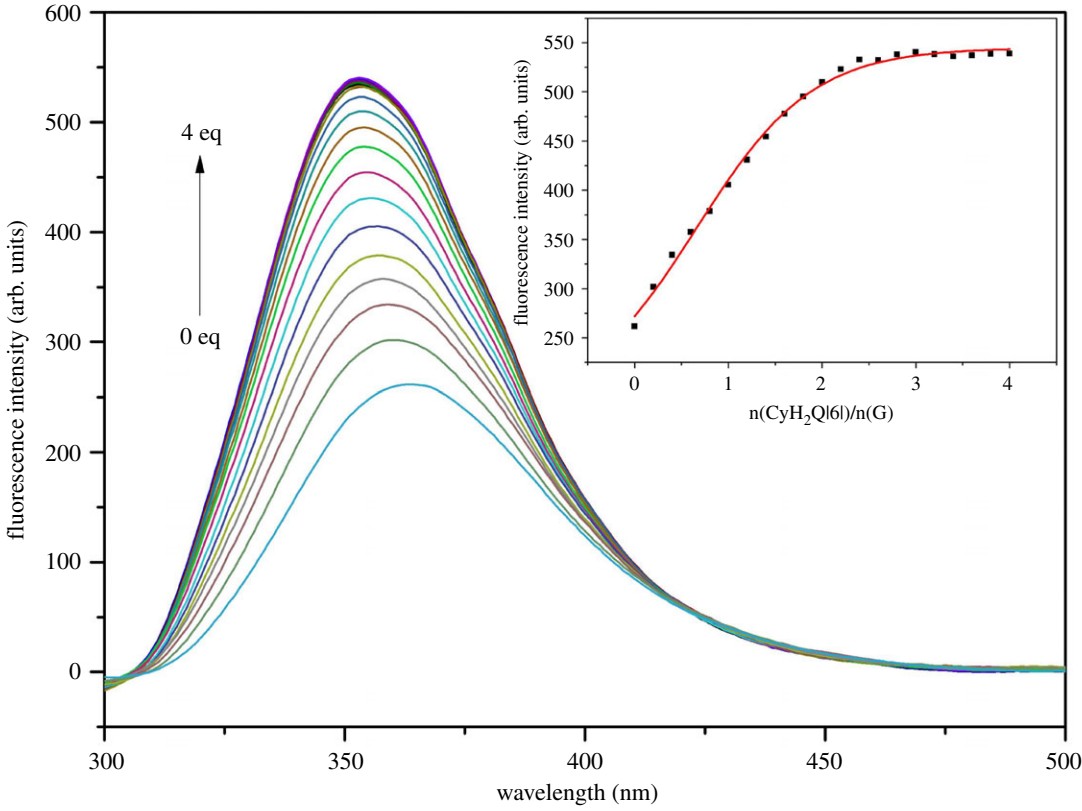

**Figure 5.** Fluorescence titration spectra of the host–guest interaction between CyH$_2$Q[6] and **G**.

## 3.4. Fluorescence spectroscopy of the host–guest interaction

Figure 5 shows the fluorescence spectra obtained for the interaction between CyH$_2$Q[6] and **G** at the maximum fluorescence emission wavelength of 363.07 nm and maximum absorption wavelength of 295 nm. Three millilitres of a $5 \times 10^{-5}$ mol l$^{-1}$ solution of **G** was added to a quartz cuvette and CyH$_2$Q[6] slowly added to the guest solution at the following ratios: (CyH$_2$Q[6])/n(G) = 0.2, 0.4,

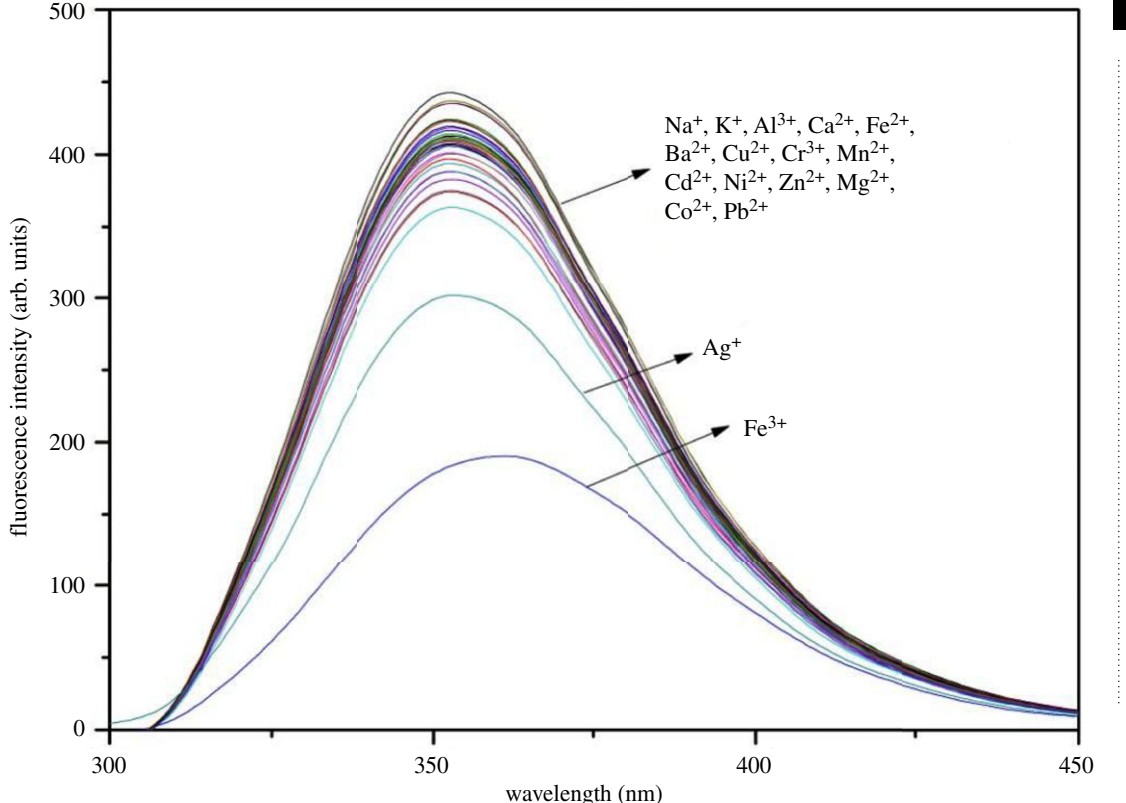

**Figure 6.** The fluorescence spectra of the **G**@CyH$_2$Q[6] ($5 \times 10^{-5}$ mol l$^{-1}$) probe in the presence of various metal cations (10 eq) recorded in an acidic aqueous solution (pH = 1).

$0.6, \ldots, 4.0$. The UV spectra indicate that the fluorescence intensity gradually increases, but upon reaching n(CyH$_2$Q[6])/n(G) = 2 : 1, the fluorescence intensity only gradually changes.

## 3.5. Determination of the metal cations selectivity using fluorescence spectroscopy

The selectivity of **G**@CyH$_2$Q[6] toward common metal cations was monitored using fluorescence spectroscopy (figure 6). In the fluorescence range of 600 nm, the **G**@CyH$_2$Q[6] probe exhibits a strong fluorescence intensity. Upon adding ten equivalents of different metal cations (Hg$^{2+}$, Ca$^{2+}$, Na$^+$, Mg$^{2+}$, Al$^{3+}$, Cd$^{2+}$, Cu$^{2+}$, Pb$^{2+}$, Ni$^{2+}$, Co$^{2+}$, Cr$^{3+}$, …), most of the metal cations do not affect the fluorescence intensity of the host and guest complex. However, Fe$^{3+}$ ions cause a significant decrease in the fluorescence intensity, which indicates that the probe was highly selective toward Fe$^{3+}$ ions.

## 3.6. Investigation of the interference of metal cations on the probe using florescence spectroscopy

Figure 7 shows the interference of other metal cations on the probe's recognition of Fe$^{3+}$. The specific operation of this experiment was as follows: 3 ml of a $5 \times 10^{-5}$ mol l$^{-1}$ solution of **G**@CyH$_2$Q[6] and Fe$^{3+}$ was added to a quartz cuvette and its fluorescence intensity measured. Different metal cation solutions (the concentration of the metal cations was 10 times that of the probe) were then added to compare whether the fluorescence intensity changes upon adding the different metal cations. The black bars in the figure shows the fluorescence intensity of the **G**@CyH$_2$Q[6] and Fe$^{3+}$ solution at an excitation wavelength of 298 nm and the red bars indicate the fluorescence intensity of the **G**@CyH$_2$Q[6] and Fe$^{3+}$ solution upon the addition of other metal cations. It can be seen from the figure that after adding other metal cations to the **G**@CyH$_2$Q[6] and Fe$^{3+}$ solution, the fluorescence intensity was changed, but the degree of change was not large. This fully shows that the Ag$^+$, Tb$^{3+}$, Er$^{3+}$ and Hg$^{2+}$ ions have weak interference to the recognition of Fe$^{3+}$, and other metal cations cannot interfere with the recognition of Fe$^{3+}$ by the complex.

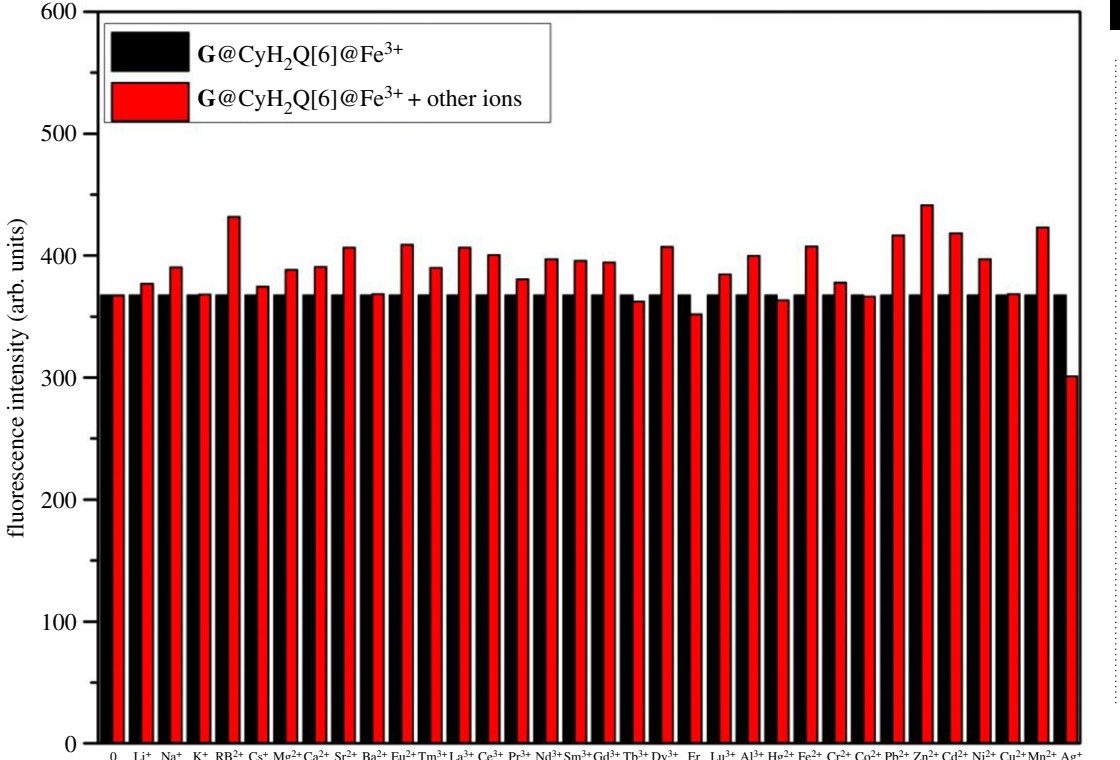

**Figure 7.** The change in the fluorescence intensity after adding different metal ions to the **G**@CyH$_2$Q[6] and Fe$^{3+}$ system.

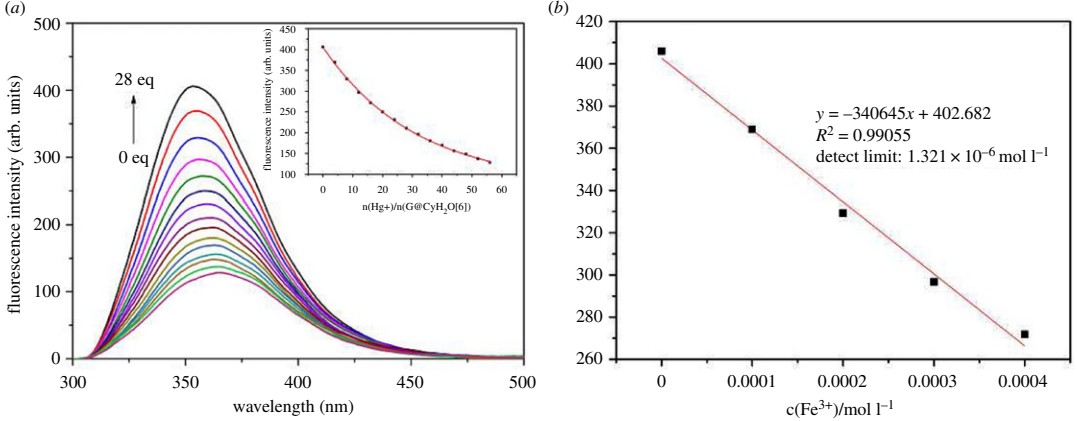

**Figure 8.** Fluorescence titration of Fe$^{3+}$ and **G**@CyH$_2$Q[6] (a) and fluorescence intensity calibration curve (b).

## 3.7. Limit of detection for the Fe$^{3+}$ titration detection probe

As shown in figure 8, the concentration of the fixed probe **G**@CyH$_2$Q[6] is $5 \times 10^{-5}$ mol l$^{-1}$, and the Fe$^{3+}$ solution of 0.2 molar ratio is added dropwise in turn. With the increase of the concentration of Fe$^{3+}$ added, the fluorescence intensity gradually decreases. According to the detection limit formula ($3\sigma/K$), the detection limit of **G**@CyH$_2$Q[6] for Fe$^{3+}$ is calculated, and the detection limit is $1.321 \times 10^{-6}$ mol l$^{-1}$, and the linear correlation is $R^2 = 0.99055$. It is lower than the maximum Fe$^{3+}$ content of 0.3 mg l$^{-1}$ specified in the national drinking water sanitation standard (GB5749-2006), so the probe **G**@CyH$_2$Q[6] can effectively identify Fe$^{3+}$ in drinking water.

## 4. Conclusion

In this paper, 2-phenylbenzimidazole was selected as the guest molecule and CyH$_2$Q[6] was used as the host to form an inclusion complex, which was used as a fluorescent probe to identify metal cations.

Firstly, the interaction between the $CyH_2Q[6]$ host and 2-phenylbenzimidazole guest in the liquid phase was studied using NMR, UV and fluorescence spectroscopy. The experimental results showed that $CyH_2Q[6]$ and 2-phenylbenzimidazole formed a 2:1 inclusion compound. There are two modes of action under acidic conditions (pH = 1). One mode is the entry of the benzimidazole into the cavity of the cucurbit[n]uril and the portal interacts with the benzene ring. The other mode is that the benzene ring enters into the cavity of the cucurbit[n]uril and the benzimidazole interacts with the portal. The interaction between $CyH_2Q[6]$ and 2-phenylbenzimidazole in the solid phase was studied using single crystal X-ray diffraction. The crystal structure results showed that $CyH_2Q[6]$ forms a 1:3 inclusion compound with 2-phenylbenzimidazole. The recognition of metal cations between $CyH_2Q[6]$ and 2-phenylbenzimidazole was studied and the fluorescent probe ($G@CyH_2Q[6]$) was constructed under acidic conditions (pH = 1). The recognition ability of the probe toward metal cations has been studied. The $G@CyH_2Q[6]$ probe can selectively identify $Fe^{3+}$ ions with a limit of detection of $1.321 \times 10^{-6}$ mol l$^{-1}$ and has a strong ability to resist cation interference.

Data accessibility. Data available from: https://www.ccdc.cam.ac.uk/data_request/cif.

Authors' contributions. J.Z. and L.A.: conceptualization, methodology, software, data curation, investigation, writing—original draft; J.G. and L.Z.: validation, formal analysis, visualization; X.Y., W.Z. and S.C.: resources, writing—review, editing, supervision, data curation; P.M.: writing—review and editing. All authors gave final approval for publication. All authors gave final approval for publication and agreed to be held accountable for the work performed therein.

Competing interests. We declare we have no competing interests.

Funding. This work was supported by the National Natural Science Foundation of China (grant no. 21762011) and Guizhou Science and Technology Planning Project (Guizhou Science and Technology Cooperation Platform Talent [2017]5788).

Acknowledgements. We thank the experts of the Key Laboratory of Macrocyclic Chemistry and Supramolecular Chemistry of Guizhou University for their technical guidance. We also thank several anonymous reviewers and editors who provided comments that greatly improved the manuscript.

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
