## [Peer Review File · Royal Society Open Science]

Review History

RSOS-211280.R0 (Original submission)

Review form: Reviewer 1

Is the manuscript scientifically sound in its present form?

No

Are the interpretations and conclusions justified by the results?

No

Is the language acceptable?

No

Do you have any ethical concerns with this paper?

Yes

Have you any concerns about statistical analyses in this paper?

No

Recommendation?

Reject

Comments to the Author(s)

In this manuscript the authors have studied a cucurbituril derivative with the chemical 2-phenylbenzimidazole via a number of chemical spectroscopic techniques. The host-guest complex has also been studied with a range of metal cations. I have serious issues with some of the conclusions drawn by the authors which is not supported by the data and the manuscript should be rejected until they are resolved.

1. The English grammar is insufficient throughout the manuscript. It requires professional proof reading before it will be at an acceptable standard.
2. In the abstract the authors state that "other cations basically do not interfere with the host-guest complex". This claim is not correct as the silver ions reduced fluorescent intensity by around 33%.
3. The introduction is too short and insufficient. Why was 2-phenylbenzimidazole selected and not a different benzimidazole? Why CyH₂Q[6] and not another cucurbituril derivative?
4. CyH₂Q[6] has been defined in the abstract but not in the main manuscript. It needs to be defined in the introduction.
5. The statement that "cucurbit[n]uril itself can only be dissolved in solutions of formic acid, concentrated acid, and concentrated alkali, and consequently, the development of cucurbit[n]urils has been significantly limited." is not correct. There are many cucurbiturils that are soluble in pure water and most cucurbiturils can be dissolved in salt solutions not just acids and alkalis.
6. Figure 1. The bond angle between the phenyl and the benzimidazole group is not correct.
7. Section 2.2. The NMR solvent is deuterium chloride not deuterium. Deuterium is just the hydrogen atoms.
8. The NMR machine details provided are insufficient.
9. The fluorescent spectroscopy details (such as excitation and emission wavelength) are missing in sections and repeated in other sections. There should be just one experimental section that gives the fluorescent machine details and operating conditions.
10. Section 2.4 The statement that the X-ray file has been uploaded to the CDCC is incorrect. A search of the site can not find the stated CDCC number.
11. Section 2.5. What is "secondary" water? This is not a recognised chemistry term.
12. Throughout the manuscript it would make it easier to read if the concentrations of solutions were given in millimolar (mM) or micromolar (uM) instead of the Molar units used (e.g. 5×10^{-5} M).
13. Section 2.6 This section should be titled "Fluorescent quenching" not "interference".

14. Figure 2. The signal-to-noise ratio in the spectra is too low to properly visualise the benzimidazole/phenyl peaks. Experimentals should be re-run with more scans to generate better spectra.
15. How were guest peaks assigned after binding by CyH₂Q[6]? How do the authors know the H₃ and H₄ protons shifted upfield and downfield? Details are needed for how these assignments were made to know if they are correct.
16. The authors are confusing 2:1 binding with two site binding. It is possible that the binding is 1:1 and that the macrocycle shifts between two locations on the guest.
17. The UV-Vis spectra are not sufficient to demonstrate 2:1 binding. A Job Plot is needed to show this.
18. Page 12. The stated fluorescent maximum and UV absorbance maximum are around the wrong ways. The numbers should be swapped.
19. Figure 5 should be changed to show possible 2:1 binding, unless the crystal structure does not show that. The right side of the figure is too complex and low resolution to show anything useful.
20. Section 3.4 should be before section 3.3. All the fluorescent sections should come one after the other.

Review form: Reviewer 2

Is the manuscript scientifically sound in its present form?

Yes

Are the interpretations and conclusions justified by the results?

Yes

Is the language acceptable?

Yes

Do you have any ethical concerns with this paper?

No

Have you any concerns about statistical analyses in this paper?

No

Recommendation?

Accept with minor revision (please list in comments)

Comments to the Author(s)

Comments

Spelling In 51 page 3

Exp section 2.2

Vagaries and inconsistencies in the experimental eg a certain amount, deuterium what?

inconsistencies throughout the text eg Fifty microlitres of this solution and 450 μ L ...,

Reportaled?

secondary water?

Fig 2 could be improved by increase the height of the relevant peaks and deleting or covering the solvent peak.

How where the shifted peak assignments determined?

Ln 20 page correspond ... relative to

Fig 5 difficult to see what is happening in c this needs to be clearer.

Figure 7 given that the black bar is unchanged why not just draw a line at that level. The aesthetics would be better.

Ln 28-30 Pg 16 ... have a weak effect on the probe, they cannot interfere with its recognition of Fe³⁺... it's a contradiction

At no time in the manuscript do the authors identify specifically which disubstituted Q[6] is being use. It is not necessarily common knowledge.

Review form: Reviewer 3

Is the manuscript scientifically sound in its present form?

Yes

Are the interpretations and conclusions justified by the results?

Yes

Is the language acceptable?

No

Do you have any ethical concerns with this paper?

No

Have you any concerns about statistical analyses in this paper?

No

Recommendation?

Accept with minor revision (please list in comments)

Comments to the Author(s)

The authors prepared and characterized new host-guest complexes for enhancing detection limit of metal ions. The concept is not novel and has been previously reported. But the selected materials are new for that purpose. This is a good expansion to the application of CB and its derivative in chemical sensing, which worth publication. work was well done, but not well written.

I have few comments that require author's response or feedback.

1. pH =1 is too acidic solution for NMR HG titration. H⁺ could compete with guest interaction and lower binding constant. Did the author try higher pH?
2. Cl⁻ also significantly quenches PBz emission and interfere with LOD results. This needs to be checked and mentioned in discussion.
3. Why was, in NMR titration, guest concentration not kept fixed by analogy to optical titration?

4. Why would sensing metal ions under these too acidic solution (pH=1) have any potential applications in water research? Have the authors tried pH of 7?
5. In figure 3, what's the red solid line representing ? Is this a 2:1 binding model function? If yes, please add to experimental section
6. It is documented that BZ binds to Hg and Rh (check ref Costa et al 19th Int Electron Conf Synth Org Chem) October 2015
Recognition of transition metals by BZ with an optical response
Why didn't authors try those metal ions?
7. Many refs are missing on CB-induced to sensing for metal ions such as
 1. Pang et al, Chem commun 201,46, 4073-4075
 2. Wei et al, Supramolecular Chem 28: 784-791
 3. Saleh et al, Michrochimica Acta 2020, 187, 386.authors need to cite all that.

Decision letter (RSOS-211280.R0)

Dear Dr Ma:

Title: Study on the host-guest complex of dicyclohexanocucurbit[6]uril and 2-phenylbenzimidazole, and its recognition effect toward Fe³⁺
Manuscript ID: RSOS-211280

The editor assigned to your manuscript has now received comments from reviewers. We would like you to revise your paper in accordance with the referee and Subject Editor suggestions which can be found below (not including confidential reports to the Editor). Please note this decision does not guarantee eventual acceptance.

Please submit your revised paper before 13-Oct-2021. Please note that the revision deadline will expire at 00.00am on this date. If we do not hear from you within this time then it will be assumed that the paper has been withdrawn. In exceptional circumstances, extensions may be possible if agreed with the Editorial Office in advance. We do not allow multiple rounds of revision so we urge you to make every effort to fully address all of the comments at this stage. If deemed necessary by the Editors, your manuscript will be sent back to one or more of the original reviewers for assessment. If the original reviewers are not available we may invite new reviewers.

When submitting your revised manuscript, you must respond to the comments made by the referees and upload a file "Response to Referees" in "Section 6 - File Upload". Please use this to document how you have responded to the comments, and the adjustments you have made. In

order to expedite the processing of the revised manuscript, please be as specific as possible in your response.

Yours sincerely,
Dr Ellis Wilde
Publishing Editor, Journals

RSC Associate Editor
Comments to the Author:
(There are no comments.)

RSC Subject Editor
Comments to the Author:
(There are no comments.)

Reviewers' Comments to Author:

Reviewer: 1

Comments to the Author(s)

In this manuscript the authors have studied a cucurbituril derivative with the chemical 2-phenylbenzimidazole via a number of chemical spectroscopic techniques. The host-guest complex has also been studied with a range of metal cations. I have serious issues with some of the conclusions drawn by the authors which is not supported by the data and the manuscript should be rejected until they are resolved.

1. The English grammar is insufficient throughout the manuscript. It requires professional proof reading before it will be at an acceptable standard.
2. In the abstract the authors state that "other cations basically do not interfere with the host-guest complex". This claim is not correct as the silver ions reduced fluorescent intensity by around 33%.
3. The introduction is too short and insufficient. Why was 2-phenylbenzimidazole selected and not a different benzimidazole? Why CyH₂Q[6] and not another cucurbituril derivative?
4. CyH₂Q[6] has been defined in the abstract but not in the main manuscript. It needs to be defined in the introduction.

5. The statement that "cucurbit[n]uril itself can only be dissolved in solutions of formic acid, concentrated acid, and concentrated alkali, and consequently, the development of cucurbit[n]urils has been significantly limited." is not correct. There are many cucurbiturils that are soluble in pure water and most cucurbiturils can be dissolved in salt solutions not just acids and alkalis.
6. Figure 1. The bond angle between the phenyl and the benzimidazole group is not correct.
7. Section 2.2. The NMR solvent is deuterium chloride not deuterium. Deuterium is just the hydrogen atoms.
8. The NMR machine details provided are insufficient.
9. The fluorescent spectroscopy details (such as excitation and emission wavelength) are missing in sections and repeated in other sections. There should be just one experimental section that gives the fluorescent machine details and operating conditions.
10. Section 2.4 The statement that the X-ray file has been uploaded to the CDCC is incorrect. A search of the site can not find the stated CDCC number.
11. Section 2.5. What is "secondary" water? This is not a recognised chemistry term.
12. Throughout the manuscript it would make it easier to read if the concentrations of solutions were given in millimolar (mM) or micromolar (μM) instead of the Molar units used (e.g. 5×10^{-5} M).
13. Section 2.6 This section should be titled "Fluorescent quenching" not "interference".
14. Figure 2. The signal-to-noise ratio in the spectra is too low to properly visualise the benzimidazole/phenyl peaks. Experimentals should be re-run with more scans to generate better spectra.
15. How were guest peaks assigned after binding by $\text{CyH}_2\text{Q}[6]$? How do the authors know the H3 and H4 protons shifted upfield and downfield? Details are needed for how these assignments were made to know if they are correct.
16. The authors are confusing 2:1 binding with two site binding. It is possible that the binding is 1:1 and that the macrocycle shifts between two locations on the guest.
17. The UV-Vis spectra are not sufficient to demonstrate 2:1 binding. A Job Plot is needed to show this.
18. Page 12. The stated fluorescent maximum and UV absorbance maximum are around the wrong ways. The numbers should be swapped.
19. Figure 5 should be changed to show possible 2:1 binding, unless the crystal structure does not show that. The right side of the figure is too complex and low resolution to show anything useful.
20. Section 3.4 should be before section 3.3. All the fluorescent sections should come one after the other.

Reviewer: 2
 Comments to the Author(s)
 Comments

Spelling ln 51 page 3

Exp section 2.2

Vagaries and inconsistencies in the experimental eg a certain amount, deuterium what? inconsistencies throughout the text eg Fifty microlitres of this solution and 450 μ L ..., Reported?

secondary water?

Fig 2 could be improved by increase the height of the relevant peaks and deleting or covering the solvent peak.

How where the shifted peak assignments determined?

Ln 20 page correspond ... relative to

Fig 5 difficult to see what is happening in c this needs to be clearer.

Figure 7 given that the black bar is unchanged why not just draw a line at that level. The aesthetics would be better.

Ln 28-30 Pg 16 ... have a weak effect on the probe, they cannot interfere with its recognition of Fe³⁺... it's a contradiction

At no time in the manuscript do the authors identify specifically which disubstituted Q[6] is being use. It is not necessarily common knowledge.

Reviewer: 3

Comments to the Author(s)

The authors prepared and characterized new host-guest complexes for enhancing detection limit of metal ions. The concept is not novel and has been previously reported. But the selected materials are new for that purpose. This is a good expansion to the application of CB and its derivative in chemical sensing, which worth publication. work was well done, but not well written.

I have few comments that require author's response or feedback.

1. pH =1 is too acidic solution for NMR HG titration. H⁺ could compete with guest interaction and lower binding constant. Did the author try higher pH?
2. Cl⁻ also significantly quenches PBz emission and interfere with LOD results. This needs to be checked and mentioned in discussion.
3. Why was, in NMR titration, guest concentration not kept fixed by analogy to optical titration?
4. Why would sensing mental ions under these too acidic solution (pH=1) have any potential applications in water research? Have the authors tried pH of 7?
5. In figure 3, what's the red solid line representing ? Is this a 2:1 binding model function? If yes, please add to experimental section
6. It is documented that BZ binds to Hg and Rh (check ref Costa et al 19th Int Electron Conf Synth Org Chem) October 2015

Recognition of transition metals by BZ with an optical response

Why didn't authors try those metal ions?

7. Many refs are missing on CB-induced to sensing for metal ions such as

1. Pang et al, Chem commun 201,46, 4073-4075
2. Wei et al, Supramolecular Chem 28: 784-791

3. Saleh et al, *Microchimica Acta* 2020, 187, 386.
authors need to cite all that.

Author's Response to Decision Letter for (RSOS-211280.R0)

See Appendix A.

RSOS-211280.R1 (Revision)

Review form: Reviewer 2

Is the manuscript scientifically sound in its present form?

Yes

Are the interpretations and conclusions justified by the results?

Yes

Is the language acceptable?

Yes

Do you have any ethical concerns with this paper?

No

Have you any concerns about statistical analyses in this paper?

No

Recommendation?

Accept with minor revision (please list in comments)

Comments to the Author(s)

Fig 2.

The authors have missed the point of improving the NMR spectra. The spectra are slightly improved by running again but running again was not the intention. The suggestion was to increase the intensity of the peaks by raising the peak heights for the relevant signals, to at least half of the space between each spectrum's baseline.

The question as to how the shifted resonances were assigned was not answered. If they were based upon shifts alone that's not necessarily valid, but only a reasonable assumption. The method should be specified.

Page 29 line 17 ... can't interfere with the recognition of Fe³⁺ ...

This statement appears to this reviewer to be factually incorrect in the context of applying this method to the detection Fe³⁺ given that 25 of the ions indicated in Fig 7 result in an enhancement of fluorescence and 4 of them cause a small decrease relative to Fe³⁺ alone.

While this could be accommodated in an analysis for Fe³⁺ with correction controls it just isn't possible to say that there is no interference.

Review form: Reviewer 3

Is the manuscript scientifically sound in its present form?

Yes

Are the interpretations and conclusions justified by the results?

Yes

Is the language acceptable?

Yes

Do you have any ethical concerns with this paper?

No

Have you any concerns about statistical analyses in this paper?

No

Recommendation?

Accept as is

Comments to the Author(s)

Thank you for your response.

Decision letter (RSOS-211280.R1)

Dear Dr Ma:

Title: Study on the host-guest complex of dicyclohexanocucurbit[6]uril and 2-phenylbenzimidazole, and its recognition effect toward Fe³⁺
Manuscript ID: RSOS-211280.R1

Thank you for submitting the above manuscript to Royal Society Open Science. On behalf of the Editors and the Royal Society of Chemistry, I am pleased to inform you that your manuscript will be accepted for publication in Royal Society Open Science subject to minor revision in accordance with the referee suggestions. Please find the reviewers' comments at the end of this email.

The reviewers and handling editors have recommended publication, but also suggest some minor revisions to your manuscript. Therefore, I invite you to respond to the comments and revise your manuscript.

Please also include the following statements alongside the other end statements. As we cannot publish your manuscript without these end statements included, if you feel that a given heading is not relevant to your paper, please nevertheless include the heading and explicitly state that it is not relevant to your work. We have included a screenshot example of the end statements for reference.

- Ethics statement

Please clarify whether you received ethical approval from a local ethics committee to carry out your study. If so please include details of this, including the name of the committee that gave consent in a Research Ethics section after your main text. Please also clarify whether you received informed consent for the participants to participate in the study and state this in your Research Ethics section.

OR

Please clarify whether you obtained the necessary licences and approvals from your institutional animal ethics committee before conducting your research. Please provide details of these licences and approvals in an Animal Ethics section after your main text.

OR

Please clarify whether you obtained the appropriate permissions and licences to conduct the fieldwork detailed in your study. Please provide details of these in your methods section.

- Data accessibility

It is a condition of publication that you make available the data and research materials supporting the results in the article. Datasets should be deposited in an appropriate publicly available repository and details of the associated accession number, link or DOI to the datasets must be included in the Data Accessibility section of the article (<https://royalsocietypublishing.org/rsos/for-authors#question17>). Reference(s) to datasets should also be included in the reference list of the article with DOIs (where available).

Please include a Data Availability section after your main text stating where supporting data are available from, or where they will be made available should your article be accepted for publication.

If you wish to submit your supporting data or code to Dryad (<http://datadryad.org/>), or modify your current submission to dryad, please use the following link:
<http://datadryad.org/submit?journalID=RSOS&manu=RSOS-211280.R1>

- Competing interests

Please include a Competing Interests section after your main text declaring any financial or non-financial competing interests. If you have no competing interests please state 'I/we have no competing interests.'

- Authors' contributions

Please include an Authors' Contributions section at the end of your main text detailing the contribution of each author. All authors should have read and approved the manuscript before submission and this should be stated in the Authors' Contributions section.

The list of Authors should meet all of the following criteria; 1) substantial contributions to conception and design, or acquisition of data, or analysis and interpretation of data; 2) drafting the article or revising it critically for important intellectual content; and 3) final approval of the version to be published.

AB carried out the molecular lab work, participated in data analysis, carried out sequence alignments, participated in the design of the study and drafted the manuscript; CD carried out the statistical analyses; EF collected field data; GH conceived of the study, designed the study,

coordinated the study and helped draft the manuscript. All authors gave final approval for publication.

- Acknowledgements

- Funding statement

Please include a funding section after your main text which lists the source of funding for each author.

Because the schedule for publication is very tight, it is a condition of publication that you submit the revised version of your manuscript before 13-Nov-2021. Please note that the revision deadline will expire at 00.00am on this date. If you do not think you will be able to meet this date please let me know immediately.

Kind regards,
Dr Ellis Wilde
Publishing Editor, Journals

RSC Associate Editor
Comments to the Author:
(There are no comments.)

RSC Subject Editor
Comments to the Author:
(There are no comments.)

Reviewer comments to Author:
Reviewer: 3
Comments to the Author(s)
Thank you for your response.

Reviewer: 2
Comments to the Author(s)
Fig 2.

The authors have missed the point of improving the NMR spectra. The spectra are slightly improved by running again but running again was not the intention. The suggestion was to increase the intensity of the peaks by raising the peak heights for the relevant signals, to at least half of the space between each spectrum's baseline.

The question as to how the shifted resonances were assigned was not answered. If they were based upon shifts alone that's not necessarily valid, but only a reasonable assumption. The method should be specified.

Page 29 line 17 ... can't interfere with the recognition of Fe³⁺ ...

This statement appears to this reviewer to be factually incorrect in the context of applying this method to the detection Fe³⁺ given that 25 of the ions indicated in Fig 7 result in an enhancement of fluorescence and 4 of them cause a small decrease relative to Fe³⁺ alone.

While this could be accommodated in an analysis for Fe³⁺ with correction controls it just isn't possible to say that there is no interference.

Author's Response to Decision Letter for (RSOS-211280.R1)

See Appendix B.

Decision letter (RSOS-211280.R2)

Dear Dr Ma:

Title: Study on the host-guest complex of dicyclohexanocucurbit[6]uril and 2-phenylbenzimidazole, and its recognition effect toward Fe³⁺
Manuscript ID: RSOS-211280.R2

It is a pleasure to accept your manuscript in its current form for publication in Royal Society Open Science. The chemistry content of Royal Society Open Science is published in collaboration with the Royal Society of Chemistry.

Yours sincerely,
Dr Ellis Wilde
Publishing Editor, Journals

RSC Associate Editor
Comments to the Author:
(There are no comments.)

Reviewer(s)' Comments to Author:

Appendix A

Title: Study on the host-guest complex of dicyclohexanocucurbit[6]uril and 2-phenylbenzimidazole, and its recognition effect toward Fe³⁺

Manuscript ID: RSOS-211280

Reviewers' Comments to Author:

Reviewer: 1

Comments to the Author(s)

In this manuscript the authors have studied a cucurbituril derivative with the chemical 2-phenylbenzimidazole via a number of chemical spectroscopic techniques. The host-guest complex has also been studied with a range of metal cations. I have serious issues with some of the conclusions drawn by the authors which is not supported by the data and the manuscript should be rejected until they are resolved.

1. The English grammar is insufficient throughout the manuscript. It requires professional proof reading before it will be at an acceptable standard.

Respond: Thank you for your suggestions. We have made necessary corrections for the grammatical errors in the manuscript and invited professional English proofreading.

2. In the abstract the authors state that "other cations basically do not interfere with the host-guest complex". This claim is not correct as the silver ions reduced fluorescent intensity by around 33%.

Respond: Thanks for your suggestion, this description has been deleted in

the abstract section.

3. The introduction is too short and insufficient. Why was 2-phenylbenzimidazole selected and not a different benzimidazole? Why CyH₂Q[6] and not another cucurbituril derivative?

Respond: Thank you for your suggestion. First of all, we think you have made a very useful suggestion. We selected for the research content is CyH₂Q[6], we will cover other modified cucurbituril in the next step work. Secondly, the same is true for choice 2-phenylbenzimidazole. In the follow-up research, we will present better works.

4. CyH₂Q[6] has been defined in the abstract but not in the main manuscript. It needs to be defined in the introduction.

Respond: A good suggestion at this time, we have defined CyH₂Q[6] in the introduction and highlighted it in yellow.

5. The statement that "cucurbit[n]uril itself can only be dissolved in solutions of formic acid, concentrated acid, and concentrated alkali, and consequently, the development of cucurbit[n]urils has been significantly limited." is not correct. There are many cucurbiturils that are soluble in pure water and most cucurbiturils can be dissolved in salt solutions not just acids and alkalis.

Respond: Thank you for your suggestion. We are deeply sorry for such an incorrect description. In the manuscript, we have modified this part and highlighted it in yellow.

"most ordinary cucurbit[n]uril have poor solubility, the development of cucurbit[n]urils has been significantly limited."

6. Figure 1. The bond angle between the phenyl and the benzimidazole group is not correct.

Respond: Thanks for your suggestion, we have changed it in Figure 1.

Figure 1. The structure of (a, b) CyH₂Q[6] and (c) 2-phenylbenzimidazole.

7. Section 2.2. The NMR solvent is deuterium chloride not deuterium.

Deuterium is just the hydrogen atoms.

Respond: Thank you for this very good suggestion, which is of great help to our work. Based on your suggestion, we have corrected this part of the error in the manuscript and highlighted it in yellow.

8. The NMR machine details provided are insufficient.

Respond: Thank you for this very good suggestion, in the experimental

part of the manuscript, we have added NMR machine details and marked it with a yellow highlight.

9. The fluorescent spectroscopy details (such as excitation and emission wavelength) are missing in sections and repeated in other sections. There should be just one experimental section that gives the fluorescent machine details and operating conditions.

Respond: Thank you for your proposal, however, the fluorescence part is separated from the laboratory, and each experiment is independent. We have already given information about the fluorescence instrument in the experimental section.

10. Section 2.4 The statement that the X-ray file has been uploaded to the CDCC is incorrect. A search of the site can not find the stated CDCC number.

Respond: Thank you for your question. We have corrected the website and can connect to CCDC through the website. http://www.ccdc.cam.ac.uk/data_request/cif.

11. Section 2.5. What is "secondary" water? This is not a recognised chemistry term.

Respond: We apologize for such an error, we have changed "secondary" to "deionized" in the manuscript.

12. Throughout the manuscript it would make it easier to read if the concentrations of solutions were given in millimolar (mM) or micromolar

(uM) instead of the Molar units used (e.g. 5×10^{-5} M).

Respond: Thank you for your suggestion, but we do this to unify the units in the manuscript, so that it will appear more rigorous.

13. Section 2.6 This section should be titled "Fluorescent quenching" not "interference".

Respond: Thank you for this very good suggestion. Section 2.6 This section have titled "Fluorescent quenching" and marked it with a yellow highlight.

14. Figure 2. The signal-to-noise ratio in the spectra is too low to properly visualise the benzimidazole/phenyl peaks. Experimentals should be re-run with more scans to generate better spectra.

Respond: Thank you for your suggestion. We tried to re-run the experiment with more scans, and the NMR spectrum we got was basically the same as that in the previous manuscript. The result is shown in the figure below.

15. How were guest peaks assigned after binding by $\text{CyH}_2\text{Q}[6]$? How do the authors know the H3 and H4 protons shifted upfield and downfield? Details are needed for how these assignments were made to know if they are correct.

Respond: First of all, thank you very much for your question. Regarding the distribution of H3 and H4 after the combination of G and the $\text{CyH}_2\text{Q}[6]$, since the $\text{CyH}_2\text{Q}[6]$ has a large hydrophobic cavity and two carbonyl ports, when the benzene ring enters the interior of the $\text{CyH}_2\text{Q}[6]$, it will be affected by the shielding effect and de-shielding effect of $\text{CyH}_2\text{Q}[6]$.

16. The authors are confusing 2:1 binding with two site binding. It is possible that the binding is 1:1 and that the macrocycle shifts between

two locations on the guest.

Respond: Thank you for your suggestion, we are based on the experiment NMR, UV, the effect of the CyH₂Q[6] and 2-phenylbenzimidazole obtained by ultraviolet is indeed 2:1.

17. The UV-Vis spectra are not sufficient to demonstrate 2:1 binding. A Job Plot is needed to show this.

Respond: Thank you for your suggestion, a Job Plot is shown in the figure below, the experimental results are consistent with the UV-Vis spectra.

18. Page 12. The stated fluorescent maximum and UV absorbance maximum are around the wrong ways. The numbers should be swapped.

Respond: Thank you very much for your suggestions. We have made

changes in the manuscript and highlighted them in yellow.

19. Figure 5 should be changed to show possible 2:1 binding, unless the crystal structure does not show that. The right side of the figure is too complex and low resolution to show anything useful.

Respond: This is a good suggestion. In the manuscript, we modified Figure 5 to make the picture clearer.

20. Section 3.4 should be before section 3.3. All the fluorescent sections should come one after the other.

Respond: In the manuscript, we have put Section 3.4 before Section 3.3.

Reviewer: 2

Comments to the Author(s)

Comments Spelling In 51 page 3

Respond: Thank you very much for your suggestions. There is a misspelling of a word in line 49 on page 3, we have rewritten this word as "cucurbituril".

Exp section 2.2 Vagaries and inconsistencies in the experimental eg a certain amount, deuterium what?

Respond: Thank you for your suggestion, we have corrected this part of the error in the manuscript and highlighted it in yellow.

inconsistencies throughout the text eg Fifty microlitres of this solution and 450 L ..., Reported? secondary water?

Respond: Thank you for your suggestion, we have used numbers to

express it uniformly in the manuscript ("50 μL of this solution and 450 μL of deuterium chloride solution.....") and changed all "secondary water" to deionized water.

Fig 2 could be improved by increase the height of the relevant peaks and deleting or covering the solvent peak. How where the shifted peak assignments determined?

Respond: Thank you, we will recollect the NMR titration data, and we can improved by increase the height of the relevant peaks and deleting or covering the solvent peak. We tried to re-run the experiment with more scans, and the NMR spectrum we got was basically the same as that in the previous manuscript. The result is shown in the figure below.

Ln 20 page correspond ... relative to

Respond: Thank you for pointing out the error for us. In the manuscript,

we have revised this sentence. "When 0.6 equivalents of **G** were added, the chemical shift of the proton signal peaks of **G** are the same as the chemical shifts observed after adding 1.0 equivalent and a free guest peak appears at the same time".

Fig 5 difficult to see what is happening in c this needs to be clearer.

Respond: This is a good suggestion. In the manuscript, we modified Figure 5 to make the picture clearer. When revising the manuscript, we swapped Figure 5 and Figure 4.

Figure 7 given that the black bar is unchanged why not just draw a line at that level. The aesthetics would be better.

Respond: Thanks for your suggestion, as you described, Figure 7 given that the black bar is unchanged, but we believe that such a picture can clearly show the interference of other ions.

Ln 28-30 Pg 16 ... have a weak effect on the probe, they cannot interfere with its recognition of Fe^{3+} ... it's a contradiction.

Respond: Thank you for your suggestion. For such an embarrassing description, we will change this sentence to "This fully shows that other metal cations can't interfere with the recognition of Fe^{3+} by the complex."

At no time in the manuscript do the authors identify specifically which disubstituted Q[6] is being use. It is not necessarily common knowledge.

Respond: Thank you for your suggestion. In the introduction part, we have defined disubstituted Q[6] in the manuscript and highlighted it in

yellow.

Reviewer: 3

Comments to the Author(s)

The authors prepared and characterized new host-guest complexes for enhancing detection limit of metal ions. The concept is not novel and has been previously reported. But the selected materials are new for that purpose. This is a good expansion to the application of CB and its derivative in chemical sensing, which worth publication. work was well done, but not well written. I have few comments that require author's response or feedback.

1. pH =1 is too acidic solution for NMR HG titration. H⁺ could compete with guest interaction and lower binding constant. Did the author try higher pH?

Respond: Thank you for your suggestion. The pH=1 solution used in the NMR titration experiment is the result of many attempts. The pH=1 is consistent in the subsequent fluorescence probe and ion interference experiments.

2. Cl⁻ also significantly quenches PBz emission and interfere with LOD results. This needs to be checked and mentioned in discussion.

Respond: Thank you for your suggestion, the influence of Cl⁻ in the whole process of identifying metal ions is constant. The results of interference to LOD are all consistent.

3. Why was, in NMR titration, guest concentration not kept fixed by analogy to optical titration?

Respond: Thank you for this point of view. Under normal circumstances, we can choose to keep the concentration of the ring or the guest fixed. In this experiment, we choose the concentration of the fixed ring to stay the same, with the guest (G) as the variable.

4. Why would sensing metal ions under these too acidic solution (pH=1) have any potential applications in water research? Have the authors tried pH of 7?

Respond: First, we thank you for your suggestion. It is also necessary to detect Fe^{3+} in acidic solutions, because many factories contain Fe^{3+} in acidic waste liquids. Secondly, the condition of pH=1 is an invariant in all our experiments.

5. In figure 3, what's the red solid line representing? Is this a 2:1 binding model function? If yes, please add to experimental section.

Respond: Thank you for your suggestion. In figure 3, the red solid line represents the trend graph of the UV absorption change of G with the addition of $\text{CyH}_2\text{Q}[6]$, this is a 2:1 binding model function.

6. It is documented that BZ binds to Hg and Rh (check ref Costa et al 19th Int Electron Conf Synth Org Chem) October 2015. Recognition of transition metals by BZ with an optical response. Why didn't authors try those metal ions?

Respond: Thank you for this suggestion. Hg ion is reflected in our experiment, and its response to the complex is not strong. As for Rh ions are not common in iron-associated minerals, what we detected in our experiments were some commonly used ions or ions in factory waste liquid.

7. Many refs are missing on CB-induced to sensing for metal ions such as

1. Pang et al, Chem commun 201,46, 4073-4075
2. Wei et al, Supramolecular Chem 28: 784-791
3. Saleh et al, Michrochimica Acta 2020, 187, 386.

authors need to cite all that.

Respond: Thank you for your suggestion.

[40] Y.Q. Xu, M. J. Panzner, X. P. Li, W. J. Youngs, Y. Pang. *Chemical communications (Cambridge, England)*, 2010, 46, 4073-4075.

[41] Q. X Geng, H. Cong, Z Tao, L. F. Lindoy, G Wei. *Supramolecular Chemistry*, 2016, 28, 784-791.

[42] R. H. Alzard, H. Meyer, F. Benyettou, A. Trabolsi, N. Saleh. *Microchim Acta*, 2020, 187, 386.

Appendix B

Comments to the Author(s)

Fig 2.

The authors have missed the point of improving the NMR spectra. The spectra are slightly improved by running again but running again was not the intention. The suggestion was to increase the intensity of the peaks by raising the peak heights for the relevant signals, to at least half of the space between each spectrum's baseline.

Respond: Thank you very much for your valuable suggestions, and thank you for your recognition of our work. Due to the unsatisfactory solubility of 2-phenylbenzimidazole (G), this spectrum is currently the most perfect.

The question as to how the shifted resonances were assigned was not answered. If they were based upon shifts alone that's not necessarily valid, but only a reasonable assumption. The method should be specified.

Respond: Thank you very much for your valuable suggestions. In Figure 2, the question as to how the shifted resonances were assigned. We are based on the fact that a certain part of the guest molecule enters the cavity of the cucurbit[n]uril. Due to the shielding effect and unshielding effect of the cucurbit[n]uril, the nuclear magnetic peak position shifts. For these reasons, we speculate that NMR has two modes of interaction described in the manuscript.

Page 29 line 17 ... can't interfere with the recognition of Fe^{3+} ...

This statement appears to this reviewer to be factually incorrect in the context of applying this method to the detection Fe^{3+} given that 25 of the ions indicated in Fig 7 result in an enhancement of fluorescence and 4 of them cause a small decrease relative to Fe^{3+} alone.

While this could be accommodated in an analysis for Fe^{3+} with correction controls it just isn't possible to say that there is no interference.

Respond: Thank you very much for your valuable suggestions, and thank you for your recognition of our work. We also realize that this description is incorrect, in the manuscript, we changed this part of the description. "This fully shows that the Ag^+ , Tb^{3+} , Er^{3+} and Hg^{2+} ions have weak interference to the recognition of Fe^{3+} , and other metal cations can't interfere with the recognition of Fe^{3+} by the complex."